# Emerging Trends in the Field of Inflammation and Proteinopathy in ALS/FTD Spectrum Disorder

**DOI:** 10.3390/biomedicines11061599

**Published:** 2023-05-31

**Authors:** Fabiola De Marchi, Toni Franjkic, Paride Schito, Tommaso Russo, Jerneja Nimac, Anna A. Chami, Angelica Mele, Lea Vidatic, Jasna Kriz, Jean-Pierre Julien, Gordana Apic, Robert B. Russell, Boris Rogelj, Jason R. Cannon, Marco Baralle, Federica Agosta, Silva Hecimovic, Letizia Mazzini, Emanuele Buratti, Ivana Munitic

**Affiliations:** 1Department of Neurology and ALS Centre, University of Piemonte Orientale, Maggiore Della Carità Hospital, Corso Mazzini 18, 28100 Novara, Italy; fabiola.demarchi@uniupo.it (F.D.M.); 20011892@studenti.uniupo.it (A.M.); 2Laboratory for Molecular Immunology, Department of Biotechnology, University of Rijeka, R. Matejcic 2, 51000 Rijeka, Croatia; toni.franjkic@student.uniri.hr; 3Metisox, Cambridge CB24 9NL, UK; gordana.apic@metisox.com; 4Department of Neurology & Neuropathology Unit, Institute of Experimental Neurology (INSPE), Division of Neuroscience, IRCCS San Raffaele Scientific Institute, 20132 Milan, Italy; schito.paride@hsr.it (P.S.); russo.tommaso@hsr.it (T.R.); 5Department of Biotechnology, Jozef Stefan Institute, SI-1000 Ljubljana, Slovenia; jerneja.nimac@ijs.si (J.N.); boris.rogelj@ijs.si (B.R.); 6Graduate School of Biomedicine, Faculty of Medicine, University of Ljubljana, SI-1000 Ljubljana, Slovenia; 7CERVO Research Centre, Laval University, Quebec City, QC G1J 2G3, Canada; anna.chami.1@ulaval.ca (A.A.C.); jasna.kriz@fmed.ulaval.ca (J.K.); jean-pierre.julien@fmed.ulaval.ca (J.-P.J.); 8Laboratory for Neurodegenerative Disease Research, Division of Molecular Medicine, Ruder Boskovic Institute, 10000 Zagreb, Croatia; lea.vidatic@irb.hr (L.V.); silva.hecimovic@irb.hr (S.H.); 9Cell Networks, University of Heidelberg, 69117 Heidelberg, Germany; robert.russell@bioquant.uni-heidelberg.de; 10Faculty of Chemistry and Chemical Technology, University of Ljubljana, SI-1000 Ljubljana, Slovenia; 11School of Health Sciences, Purdue University, West Lafayette, IN 47907, USA; cannonjr@purdue.edu; 12Purdue Institute for Integrative Neuroscience, Purdue University, West Lafayette, IN 47907, USA; 13RNA Biology, ICGEB, 34149 Trieste, Italy; marco.baralle@icgeb.org; 14Neuroimaging Research Unit, Institute of Experimental Neurology, Division of Neuroscience, IRCCS San Raffaele Scientific Institute, 20132 Milan, Italy; agosta.federica@hsr.it; 15International Centre for Genetic Engineering and Biotechnology (ICGEB), Padriciano 99, 34149 Trieste, Italy

**Keywords:** amyotrophic lateral sclerosis, frontotemporal degeneration, inflammation, proteinopathy, neurodegenerative diseases

## Abstract

Proteinopathy and neuroinflammation are two main hallmarks of neurodegenerative diseases. They also represent rare common events in an exceptionally broad landscape of genetic, environmental, neuropathologic, and clinical heterogeneity present in patients. Here, we aim to recount the emerging trends in amyotrophic lateral sclerosis (ALS) and frontotemporal degeneration (FTD) spectrum disorder. Our review will predominantly focus on neuroinflammation and systemic immune imbalance in ALS and FTD, which have recently been highlighted as novel therapeutic targets. A common mechanism of most ALS and ~50% of FTD patients is dysregulation of TAR DNA-binding protein 43 (TDP-43), an RNA/DNA-binding protein, which becomes depleted from the nucleus and forms cytoplasmic aggregates in neurons and glia. This, in turn, via both gain and loss of function events, alters a variety of TDP-43-mediated cellular events. Experimental attempts to target TDP-43 aggregates or manipulate crosstalk in the context of inflammation will be discussed. Targeting inflammation, and the immune system in general, is of particular interest because of the high plasticity of immune cells compared to neurons.

## 1. Introduction

The pathological mechanisms in neurodegenerative diseases are complex and not completely understood. In amyotrophic lateral sclerosis (ALS), multiple mechanisms including defects in proteostasis, RNA metabolism, nucleocytoplasmic transport, vesicle and axonal trafficking, DNA repair, and mitochondrial, oligodendrocyte and immune functions are proposed to lead to the same outcome of motor neuron death [1]. Such complexity makes it challenging to develop a single treatment that would effectively halt or reverse disease progression. Potential therapeutic targets include two main converging points in neurodegeneration: proteinopathy and inflammation. 

One of the most important discoveries on ALS pathophysiology came in 2006, when TAR DNA-binding protein 43 (TDP-43) was identified as a major component of ubiquitinated inclusions in both ALS and frontotemporal degeneration (FTD) [2,3]. TDP-43 aggregation occurs in neurons and glia of approximately 97% of ALS and ~50% of FTD patients. These two neurodegenerative diseases were traditionally regarded as distinct but are now considered as opposite ends of the same clinicopathological spectrum [4,5]. Although ALS primarily affects cortical, bulbar, and spinal motor neurons, half of the patients develop cognitive defects, with as many as 15% eventually exhibiting a severe form of FTD. Ever since the first description of TDP-43 aggregation, there has been a lively ongoing debate on whether the pathological mechanisms following this aggregation belong to a loss- or (toxic) gain-of-function scenario [6], although these scenarios are not mutually exclusive [7]. Recently, considerable attention has been focused on mapping the direct pathways affected by the depletion of nuclear TDP-43 following its cytoplasmic aggregation (Figure 1). Among these, neuroinflammation is of major interest, since protein aggregates, mitochondrial damage and/or damaged/dying neurons directly trigger microglial activation [8].

It is now widely accepted that neuroinflammation is likely one of the main causes of neuronal death, which is rarely cell autonomous [9,10,11]. Moreover, increasing evidence pinpoints the critical role of not only local central nervous system (CNS) immune cells but also the peripheral immune system in regulating ALS and FTD pathogenesis [12], strongly arguing that ALS and FTD are systemic diseases. However, the exact role played by inflammation, and the immune system in general is unclear. Currently, there is still little agreement on whether the immune response causes, worsens, or counteracts neurodegeneration [13]. Therefore, even though immune imbalance, detectable in both the CNS and in the periphery, is a common feature of all neurodegenerative diseases, and there is a variety of immunomodulatory therapies at disposal, the missing pieces in the proteinopathy and inflammation crosstalk preclude the translation of the current knowledge into disease-modifying therapies in ALS and FTD [14]. Here, we will review the underlying genetics and mechanisms in ALS/FTD spectrum disorder, and summarize its two main overlapping hallmarks—TDP-43 proteinopathy and immune system imbalance. Finally, we also provide an overview of currently available biomarkers and emerging therapies. 

## 2. ALS and FTD Genetics, Epigenetics and Mechanisms

### 2.1. Mendelian Genetic Elements 

Although most ALS and FTD cases are sporadic, known Mendelian genetic elements have been identified in ~20% of ALS [15] and ~30% of FTD patients [16]. There is a strong genetic overlap between the two diseases—currently most of the confirmed causative mutations in ALS have also been identified in FTD or ALS/FTD patients. The overlapping genes include *C9ORF72*, *TBK1*, *TARDBP*, *FUS*, *VCP*, *OPTN*, *CHCHD10*, *SQSTM1*, *TIA1*, *CCNF*, and *CYLD* (Figure 2). Other confirmed genes in ALS are *VAPB*, *EPHA4*, *UNC13A*, *NEK1*, *SOD1*, *HNRNPA1*, *ANXA11*, *UBQLN2*, *PFN1*, and *KIF5A*, whereas *GRN*, *MAPT*, and *CHMP2B* mutations have been found solely in FTD [16]. Importantly, with genome-wide association studies (GWAS) many genetic variants were found associated with these diseases. These studies will certainly increase the number of linked genes since some of the associations are recent and studies confirming the associations or identifying the functional cause are pending. In this section, we will not detail all the genetic associations and mechanisms of action identified, which can be found in several online databases [17,18]: ALSoD (http://alsod.iop.kcl.ac.uk/, accessed on 28 May 2023), ALSGene (http://www.alsgene.org, accessed on 28 May 2023), and https://www.ebi.ac.uk/gwas/home, accessed on 28 May 2023. However, we will specifically aim to highlight examples directly linked to neuroinflammation. As will be argued, these direct links strongly suggest that the immune system activation is not merely a (late) consequence of proteinopathy and/or neuronal damage or death. We will also briefly mention genes leading to autophagy failure, which due to increased proteotoxic stress and/or decreased clearance of inflammatory signaling machinery can trigger inflammatory responses as well. 

A pathogenic (G_4_C_2_)_n_ repeat expansion within the first intron of the ORF 72 on chromosome 9 (*C9ORF72*) gene is the most common mutation identified in both ALS and FTD [19,20]. Numerous cell-specific functions have been ascribed to the C9orf72 protein, including the regulation of autophagy and vesicular trafficking in neurons and myeloid cells [21]. It leads to toxic gain-of-function by two mechanisms: (1) repeat-containing RNAs transcribed from *C9ORF72* expansions sequester functionally important RNA-binding proteins in RNA foci, and (2) repeat-associated non-AUG (RAN) translation generates aggregate-prone dipeptide repeat peptides (DPR) [22]. Regarding its role in neuroinflammation, the loss-of-function, due to reduced *C9ORF72* gene expression, has been shown to have more direct consequences than gain-of-function. Loss-of-function occurs as the expansion in a noncoding part of the gene binds trimethylated histones thereby promoting heterochromatin formation and reducing transcription [23]. Mice deficient in C9orf72 show neuroinflammation with aging as well as proinflammatory phenotypes with increased expression of interleukin (IL)-6 and IL-1β, although they do not show ALS/FTD-like neuropathology [24]. A molecular mechanism between inflammation and *C9ORF72* loss-of-function has recently been identified in mice with cell-specific depletion of the gene in myeloid cells. These cells had reduced degradation of an adaptor protein that induces the secretion of type I interferons and proinflammatory cytokines, stimulator of interferon genes (STING), due to impairment of the autolysosomal pathways [25,26]. 

Mutations in the Cu/Zn-binding superoxide dismutase (*SOD1*), encoding an antioxidant enzyme that functions as a homodimer, binding copper and zinc ions, to destroy superoxide radical (O2^–^) in the body, were the first to be associated with ALS [27]. Mutations have been described as loss- and gain-of-functions. One common ALS-linked mutation in *SOD1* is the SOD1-G93A missense mutation. Macrophages and microglia from transgenic mouse models carrying the SOD1-G93A mutation are proinflammatory and secrete several proinflammatory cytokines such as tumor necrosis factor α (TNF-α), interferon γ (IFN-γ), and IL-1β, and produce more superoxide. As a result, they are more toxic to primary cultured neurons than wild-type (WT) cells [28,29]. 

Evidence of a novel regulator of inflammation has been recently gathered for optineurin, a protein that regulates multiple cellular processes, including vesicular trafficking, autophagy, inflammatory, and antiviral signaling [30]. Mutations in *OPTN* gene are linked to both ALS and FTD [31,32] and their role in inflammation has recently been reviewed [30,33,34]. Briefly, the mechanisms involved are thought to arise from loss-of-function mutations that may affect inflammation via the nuclear factor-κB (NF-κB) and/or interferon regulatory factor 3 (IRF-3) pathways [31,35,36]. However, the exact pathogenic mechanisms are still inconclusive due to discrepancies arising from different experimental models [37]. 

Mutations in the *GRN* gene have been associated with an increased risk for FTD [38,39]. This gene encodes progranulin, which has been shown to modulate the inflammatory response through various mechanisms [40]. Impaired host defense, and neuropathology, have been observed in progranulin-deficient mice [41]. The spectrum of mutations identified thus far indicates that haploinsufficiency of progranulin is the predominant mechanism of contribution to the disease [42]. 

*TANK*-*binding kinase 1* (*TBK1*) is a recently identified gene associated with ALS and FTD [43,44]. Subsequent studies indicated that *TBK1* mutations may occur frequently in patients with ALS/FTD [45]. TBK1 is a kinase involved in many different signaling pathways [46], among which, of particular interest to ALS/FTD, are those that modulate inflammation by activation of type I IFNs and proinflammatory cytokines [47] as well as autophagy [48]. The spectrum of mutations covers the whole range of possibilities. The haploinsufficiency contributes to the pathology in ALS/FTD due to loss-of-function, whereas the role of missense mutations and single amino acid deletions has yet to be determined. Functional characterization of these mutations in the context of inflammation and autophagy should be focused on to identify their specific mechanisms. These could at least in part be due to modifications of the protein binding affinity for partners, as was the case with the TBK1 E696K, which fails to bind optineurin [44]. 

A rare missense mutation in *CYLD*, a lysine-63 deubiquitinase, has identified a novel causative gene for ALS and FTD with links to neuroinflammation [49]. This enzyme acts as a negative regulator of the NF-κB pathway. ALS/FTD-linked mutation so far seems to function via a gain-of-function mechanism, as its overexpression in cell lines resulted in enhanced inhibition NF-κB and impairment of autophagosome fusion to lysosomes. 

Notably, a substantial subset of ALS and FTD patients carry mutations in the genes that encode autophagy related proteins, suggesting that autophagy contributes to proteotoxic stress in both diseases. These genes include *SOD1* and *UBQLN2* (in ALS), *CHMP2B*, *MAPT*, *GRN* and *SQSTM1* (in FTD) and *OPTN*, *C9ORF72* and *TBK1* in both ALS and FTD (Figure 2, Table 1). Different mouse and cellular models were designed to better understand how these mutations disrupt autophagy. In the mice model carrying P497S UBQNL2 mutation, increased accumulation of *SQSTM1*/p62 and ubiquitinated proteins was observed [50]. SOD1^G93A^ mice exhibited higher age-dependent accumulation of LC3-II in the spinal cords compared to WT controls, suggesting an increase in autophagy [51]. Moreover, in Neuro2A (N2A) and NSC-34 neuroblastoma/motor neuronal cell lines, L341V *SQSTM1*/p62 mutation caused defective recognition of LC3-II with impaired recruitment of SQSTM/p62 to autophagosomes [52]. In the spinal cord neurons of *OPTN*-deficient mice, diminished numbers of lower motor neurons, together with neuronal accumulation of CHMP2B positive cytoplasmic vacuoles were observed, suggesting defective autophagy [53]. Notably, some of the ALS/FTD overlapping genes (*OPTN*, *TBK1*, *C9ORF72*) play an important role both in autophagy and immune signaling and mutations in these genes were shown to disrupt both processes (Figure 2). The crosstalk between these proteins has also been reported: for example, TBK1, p62, and optineurin partner in the same molecular pathways. TBK1 is important for the phosphorylation of optineurin and p62 during autophagy, and optineurin regulates cellular localization and activation of TBK1 during inflammatory/anti-viral signaling [54,55,56,57]. The G4C2 expansion of *C9ORF72* has been reported to affect autophagy by at least three different mechanisms—dipeptide repeat aggregation, reduced endolysosomal trafficking and impaired lysosomal function, and autophagy initiation via ULK1 [58,59,60,61,62,63]. Besides this, *OPTN* and *TBK1* ALS/FTD-related mutations were shown to disrupt the selective degradation of mitochondria (mitophagy) [64]. 

### 2.2. GWAS Links

The numbers of genetic associations are much higher—the proliferation of GWAS over the last 15 years resulted in the identification of a series of variants and risk alleles, whose direct mechanism(s) or connection(s) to the pathology still need to be elucidated. For example, GWAS have postulated associations for a series of genes: *DHX58*, *TRIM21*, and *IRF7* that function within the TBK1/IRF3 immune pathway [98]. Furthermore, investigation of selected polymorphisms in genes involved in inflammation (*IL1B* rs1071676) and oxidative stress (*SOD* rs4880, CAT rs1001179) have also shown these to be possible disease modifiers [99]. As recently reviewed, further investigation into loss- and/or gain-of-function of these will provide insight into their role in the pathology, the design of therapeutic strategies, and the genetic profiling of patients [100]. One of the more recent success stories in this regard revolves around the single-nucleotide polymorphisms (SNPs) identified in *UNC13A* [101]. This example highlights the importance of analyzing the effect of SNPs that occur in a background where in most of the cases TDP-43 aggregation has occurred. Thus, the contribution of SNPs to the pathology may only be evident in a TDP-43 depleted scenario. This was in fact the case with *UNC13A* SNPs rs12608932 (A > C) and rs12973192 (C  >  G) that was identified via GWAS to be associated with ALS [101,102]. UNC13A functions in several steps involved in neurotransmission [103]. Homozygosity for the C-allele at rs12608932 has been associated with susceptibility and with shorter survival of ALS patients [104]. However, it was not until recently that studies have shown how the absence of functional TDP-43 could result in cryptic exon activation [105,106]. Briefly, the two *UNC13A* SNPs were observed to be located within an intron containing a cryptic exon that was used upon TDP-43 depletion and whose inclusion results in nonsense-mediated decay of the mRNA, and, consequently, loss of the functional protein. While the SNPs did not cause cryptic exon inclusion per se, their presence enhanced the effect caused by TDP-43 depletion. Mechanisms such as this one, which exert their action only in combination with the pathological background state, might also partly explain the effect of the other risk SNPs on disease progression.

### 2.3. Environmental Factors and Epigenetics 

As mentioned above, clear genetic linkage is present only in a fraction of patients, whereas sporadic cases comprise ~80% of cases in ALS and 70% in FTD [15]. Although major progress has been made in the field of genetic studies in the last years, they present the tip of the iceberg, with likely a large uncharted area waiting to be mapped concerning the impact of epigenetics on the development and progression of both ALS and FTD (Figure 2). So far, several epigenetic risk factors have been associated with ALS, such as intensive physical activity, exposure to pesticides and toxins, smoking, and brain injury [107]. For FTD, the most common epigenetic risk factors include alcohol overdose, smoking, exposure to pesticides and toxins, and long-term use of selenium-containing dietary supplements [108]. For both ALS and FTD, ageing is also one of the most important risk factors [109]. Notably, most of the risk factors directly affect the immune response. However, little is yet known about the exact relationship between genetics and epigenetics and their impact on both ALS and FTD. 

Using a different approach, integrating motor neuron epigenetic features with ALS GWAS data, 690 potential ALS risk genes and 36% of SNP-based heritability have been identified [110]. Another example regards the flip side of the coin, that is variants that may be protective against the pathology, as emphasized by a recent example closely connected to neuroinflammation in the interleukin 18 receptor accessory protein (IL18RAP). Variants in the 3′UTR of IL18RAP were found to be enriched in non-ALS genomes [111]. Mechanistically, these variants reduced the binding of dsRNA-binding proteins that are known to stabilize mRNAs. Consequently, IL18RAP expression and NF-κB signaling are dampened in microglia.

## 3. Proteinopathy in ALS and FTD 

Until a few years ago, the accumulation of aggregated proteins in ALS and FTD associated with TDP-43 (FTD-TDP) brains was considered to be represented solely by TDP-43. Therefore, many studies trying to better characterize the proteinopathy in ALS/FTD have focused on understanding it in terms of gain- or loss-of-function (which may include toxicity of the aggregates, altered RNA processing and transcriptional pathways, or protein–protein interaction profiles). This aspect of proteinopathy has already been analyzed in depth by several previous reviews [112,113], and will therefore not be detailed here. However, a relatively novel—currently understudied—aspect is the composition of the aggregates and the way it may affect proteinopathy. In fact, although TDP-43 certainly represents the primary aggregating protein, it is now very clear that several other cellular factors can be present in these inclusions [114]. As a result, over time, several proteins that are involved in ALS pathways and other pathogenic processes have been reported. Notably, TDP-43 pathology is not limited to ALS and FTD and it can be found in several other neurodegenerative diseases. The most significant proteins present in the aggregates are listed in Table 2.

A lot of information is lacking regarding the role played by all these co-aggregating proteins in disease pathology. Moreover, it is very likely that these proteins represent only a partial list of cellular proteins that can eventually become “trapped” in the aggregates together with the primary TDP-43 protein. In keeping with this, a previous proteomic study of insoluble material obtained from the brain of patients suggested that many proteins can potentially become recruited or trapped by TDP-43 aggregates: in one study laser capture microscopy coupled with mass-spectrometry detected altered proteins in hippocampal dentate granule cells excised from three post-mortem FTD cases. This study identified Septin 3 and 7 as potential inclusion-associated proteins in addition to TDP-43 in these samples [154]. Of course, the key issue for all these inclusion-associated proteins remains whether their function is impaired following their presence in the insoluble brain aggregates. For TDP-43, the recent discovery of cryptic exon recognition following its aggregation in neurons has clearly demonstrated a loss-of-function effect [155]. However, for all these co-aggregating proteins it is still not clear if they are sequestered to a sufficiently high extent to change their cellular functionality. Intriguingly, altered functionality has been observed for hnRNP K aggregates in a significant number of FTD-TDP cases [156]. Although these aggregates did not colocalize with TDP-43 aggregates, this finding suggests that TDP-43 might not totally account for all toxicity of disease-associated aggregates.

Another emerging concept in TDP-43 is represented by the presence of TDP-43 “seeds” within the aggregates. This is based on the observation that TDP-43 aggregates within neurons can present themselves in different shapes and their appearance is actually used to define the major FTD subtypes [157,158]. The observation that they often correlate with pathology has led to the hypothesis that different types of aggregates may possess different seeding potential. Until recently, there was little experimental support for this hypothesis, but a recent report has confirmed that different types of seeds can give rise to different types of neoaggregates [159]. Together with these types of studies, it is expected that the new advances in Cryo-EM spectroscopy will be able to shed more light on the different conformations that TDP-43 fibrils can assume within aggregates both in vitro but especially in vivo [160]. In keeping with this approach, using this technique the presence of TDP-43 filaments in ALS and FTD that adopt a unique double-spiral-shaped fold has recently been reported [161].

Finally, a still unanswered question is represented by the exact mechanism by which TDP-43 aggregates contribute to ALS. Some studies have suggested that the aggregates may be toxic to cells, especially in the case of larger inclusions or aggregate subtypes [159,162]. At least in the initial stages of the disease, however, there is still the possibility that aggregates may represent a protective response to cellular stress [163]. For certain, once the aggregates become big enough to induce TDP-43 loss-of-function in cells, it is now clear that many RNA misprocessing events can occur, which have recently been shown to include cryptic exon inclusion and which would eventually lead to extensive cytotoxicity [73,155,164,165]. Finally, it is also possible that TDP-43 aggregates may be a secondary effect of the disease rather than a primary cause. As a result, while TDP-43 aggregates are a hallmark feature of ALS, their exact role in the disease is still a subject of ongoing research and debate in the scientific community. Based on available data, it appears highly likely that the pathological aggregation of TDP-43 in the cytoplasm is a significant contributor to the development of ALS [166]. Thus, researchers are exploring various strategies for developing treatments targeting TDP-43, as will be discussed below.

## 4. Immunity in ALS and FTD 

### 4.1. Immune System in Pathogenesis of Neurodegeneration: Brief Overview of Key Evidence 

Prior to the above-discussed genetic evidence linking inflammation to neurodegeneration, multiple lines of evidence showed that the immune system is important in two opposing roles: precluding and facilitating neurodegeneration [167,168]. The primary role of the innate immune system is to rapidly protect an organism from external and internal stressors. This is carried out by microglia in the CNS and macrophages at the neuromuscular junction (NMJ). However, the protective functions break down during chronic stimulation, resulting in proinflammatory skewing. Seminal studies from the 1990s have found microglial activation, T cell infiltration and antibody accumulation in the CNS autopsies of patients with ALS, AD, and several other neurodegenerative diseases (Figure 3) [169,170,171,172,173,174]. The level of neuroinflammation, measured as microgliosis and astrocytosis, can now also be monitored to a certain extent in living patients, although still without precision [14,175]. Further insights came from immune targeting experiments in transgenic animal ALS models carrying mutated human SOD1 (mSOD1). Depending on the disease stage, activated glia can exert various anti-inflammatory (secrete cytokines IL-10 and IL-4, phagocytose debris, increase damage sensor expression) and proinflammatory functions (secrete cytokines, such as tumor necrosis factor (TNF), IL-β, IL-6, chemokines CCL2 and IL-8, increase damage sensor expressions, generate ROS and nitric oxide) (reviewed in [9,10]). These models also showed that glial and myeloid cell-specific transgene deletion or replacement of transgene-carrying microglia and macrophages with WT cells increased survival [176,177,178,179]. In contrast, T cell deficiency decreased survival in mSOD1 models, which could be recovered by the adoptive transfers of CD4 T cells, and more specifically, regulatory cells (Tregs) [180,181,182]. The latter was particularly informative because it showed that an adaptive arm of the immune system, which is present within the brain parenchyma in minute quantities, could nevertheless exert protective functions in the CNS. Follow-up studies in ALS patients showed that fast progression was linked to decreased and dysfunctional Tregs [183,184], and profiling of T cells in ALS patients’ cerebrospinal fluid (CSF) and blood has linked activated Treg numbers to longer survival [185]. These findings led to a major paradigm shift from the prevailing view that any T cell action beyond the blood–brain barrier is uniformly noxious. Moreover, it paved the way for exploring Treg therapies in ALS [186,187,188], which will be discussed below. In contrast to Tregs, the presence of activated effector CD4 T cells, cytotoxic functions and Th1 skewing were linked to poorer prognosis [185], corroborating previous findings in animal models [189]. It is of note that numerous other immune perturbations have been linked to the rate of progression in both animal models and ALS patients, such as natural killer (NK), monocyte, and CD8 T cell entry to CSF or CNS parenchyma (Figure 3) (reviewed in [14]). However, the patient immune profiling is still a limiting factor and at times conflicting because activation of the immune system is influenced by genetic and environmental factors, and it dynamically changes over the course of the disease. Notably though, the heterogeneity is very prominent in ALS in which survival may range from months to decades following an initial diagnosis, with approximately 10–20% of patients developing rapidly progressive disease and dying within the first year. Therefore, there is an urgent need for early detection of potentially unique molecular signatures associated with the more aggressive forms of the disease.

### 4.2. ALS as a Systemic Immune Disorder: Crosstalk between Immune Signaling and Metabolism

In addition to changes in the CNS, multiple studies have shown functional alterations of peripheral myeloid cells (NMJ macrophages, peripheral blood monocytes, etc.) in ALS (Figure 3) [190,191,192,193,194,195,196,197]. The gene profiling studies performed on blood monocytes from ALS patients revealed a molecular signature of chronic activation through the lipopolysaccharide (LPS)-Toll-like receptor (TLR)2/4 signaling pathway, suggesting a systemic deregulation of the innate immune response [192], although without an overt increase in blood proinflammatory markers [198,199]. Furthermore, recent studies associated altered immune profiles of myeloid cells and peripheral blood monocytes with ALS clinical features including disease severity and progression [200]. In keeping with this view, Yildiz and colleagues intriguingly showed that higher levels of activated CD11b^+^ myeloid cells in blood were linked to increased survival of ALS patients [201]. An open question is what are the underlying mechanisms that may link changes in peripheral immunity and the clinical outcome of the disease. Recently, an unbiased screening approach of 62 immune factors from the plasma of sporadic ALS patients and controls was applied to assess the effect of deregulated peripheral immunity on disease progression in ALS [198]. Surprisingly, several immune mediators (leukemia inhibitory factor (LIF), tissue inhibitor of metalloproteinase 1 (TIMP-1), tissue inhibitor of metalloproteinase 2 (TIMP-2), serum amyloid A (SAA), macrophage inflammatory protein-1 beta (MIP-1β), IFN-γ, TNF-α and monocyte chemoattractant protein (MCP-1)) were decreased in the plasma of ALS patients, coupled with a decrease in the metabolic sensor leptin. A similar molecular profile was observed in the plasma of mSOD1(G93A) mice. Therefore, data from both humans and mice suggest the presence of a shutdown of peripheral (blood) innate immune responses associated with a marked downregulation of leptin. Of note, in fast progressing patients, a differential increase in sTNF-RII and CCL16 plasma levels was associated with a more prominent decrease in plasma leptin. Similarly, ALS patients’ plasma and/or sTNF-RII led to AMP-activated protein kinase (AMPK) phosphorylation and subsequent decrease in leptin production by human adipocytes, which was more pronounced upon exposure to plasma from fast progressing patients [198]. Plasma leptin, primarily derived from adipocytes, is increased in obesity to favor satiety and energy consumption. Its role in ALS is not completely understood, but some reports have suggested that it is inversely associated with the risk of developing disease [202]. In addition, recent studies on the TDP-43 (A315T) mouse model revealed a reduction in leptin levels at the end-stage of the disease, whereas recombinant leptin supplementation improved motor performance and delayed weight loss [203,204]. Taken together, there is increasing evidence that chronic and systemic deregulation of immune response may represent one of the key elements in the pathobiology of ALS [9,10]. Importantly, distinct plasma immune profiles in sporadic disease may be associated with different clinical outcomes. Finally, increasing evidence also suggests that there is a “pathogenic crosstalk” between the immune system and metabolic signaling in sporadic ALS, but more research is needed to assess whether targeting leptin and/or the immune-metabolic axis has therapeutic potential. 

### 4.3. The Trilemma on the Origin of Immune Imbalance

Although various aspects of inflammation are detectable in all ALS and FTD patients, particularly at late disease stages, the immune-mediated pathology has been difficult to classify. A healthy immune system responds proportionally to the degree of challenge and in the best case reverts the system back to homeostasis. In disease, three different scenarios can preclude a healthy return to homeostasis: excessive inflammatory response, immunodeficiency, and autoimmunity (Figure 4). 

#### 4.3.1. Excessive Inflammatory Response 

The mechanisms that link TDP-43 proteinopathy to excessive inflammation have been discussed in several recent reviews [9,10,205]. In brief, protein aggregates arising from aggregate-prone genetic mutations or defects in proteasomal or autophagic protein disposal (age- or stress-related or linked to genetic mutations) can directly stimulate inflammation by engaging various shared receptors for damage- and pathogen-associated molecular patterns (DAMP/PAMP). This leads to proinflammatory skewing and/or inflammasome activation [206]. Interestingly, TDP-43 can also directly activate key inflammatory signaling pathways via binding RelA/p65 subunit of the transcription factor NF-κB [207,208]. In contrast, experimental TDP-43 deletion in microglia enhances phagocytosis in an AD model, thereby decreasing amyloid beta (Aβ) load but also increasing synaptic loss [209]. Notably, an interaction between proteinopathy and inflammation is bidirectional, since LPS increases TDP-43 aggregation microglia [210]. Excessive inflammatory responses are also enhanced during ageing because of immunosenescence, which profoundly affects ratios of adaptive immune cells subsets (decreases regulatory and increases effector/memory cells), and inflammageing, a proinflammatory skewing of innate immunity [211].

#### 4.3.2. Immunodeficiency 

Immunodeficiency or inefficient immune response has been proposed as a disease mechanism in neurodegeneration based on findings from experimental animal models [212,213], and more recently since ALS- and FTD-linked mutations have been found in several genes that directly regulate the innate immunity and/or are predominantly expressed in immune cells [10], as discussed above. However, due to the multifunctional properties of several of these genes, it is still hard to say if immunodeficiency is the primary trigger for neurodegeneration or whether it acts in concert with dysfunctionality in other disease mechanisms, such as, for example, autophagy for *TBK1*, *OPTN*, and *C9ORF72*. It is notable though that inefficient immune response, due to the inability to efficiently clean up the damage, eventually leads to excessive inflammation, thus resulting in the same outcome as proteinopathy itself. 

#### 4.3.3. Autoimmunity 

Although several epidemiologic studies have suggested an increased ALS risk in patients suffering from several autoimmune diseases [214], autoimmunity was considered as the least likely scenario in ALS, since contrary to multiple sclerosis (MS), there is much lower infiltration of peripheral immune cells, much more subtle breakdown of the blood–brain barrier, and ill-defined relevance of autoantibodies to CNS antigens [215,216]. In addition, GWAS studies showed no overlap between ALS and five autoimmune diseases (Crohn’s disease, ulcerative colitis, type 1 diabetes, celiac disease, and psoriasis), and minimal overlap with rheumatoid arthritis [217]. It is interesting to note that the same GWAS study showed that FTD, despite a substantial genetic and pathological overlap with ALS, had a strong genetic enrichment with all of these autoimmune diseases, in particular in the human leukocyte antigen (HLA) locus [217]. This observation could perhaps suggest that differential immune system activation may explain genetic pleiotropy, i.e., the development of either ALS or FTD in carriers of the same mutations. Nonetheless, a recent Mendelian randomization study has corroborated the finding of the absence of causality between autoimmune disorders and ALS [218], proposing that the previously found positive and negative linkage by Li et al. [219] could simply be due to genetic pleiotropy. However, although autoimmunity is unlikely to occur for most ALS cases, it has been evoked as a possibility in certain mutation carriers such as those with alterations in *C9ORF72* and senataxin (*SETX*) genes. Specifically, *SETX* mutations have been reported to cause juvenile-onset slow-progressing ALS 4 subtype [220]. It is notable that the latter has also been linked to clonally expanded effector memory CD8+ T cells in the peripheral blood, something that has previously not been reported for ALS but has been linked to AD and Parkinson’s disease (PD).

## 5. Clinical Studies, Biomarkers and Emerging Therapies in ALS/FTD 

As already described, ALS and FTD share similar pathogenic events, from the pathological deposition of misfolded proteins to progressive neuronal damage associated with altered neuroinflammatory response [221]. These shared mechanisms are of great interest since they can be exploited to develop biomarkers and therapies that act on the same patterns in different neurodegenerative diseases. 

### 5.1. Biomarkers 

Searching for disease-specific biomarkers is a pivotal step for facilitating earlier patient recruitment and selection for clinical trials. In fact, specific patient subgroups could be more likely to show a common biological effect and demonstrate target engagement, which can be used as a surrogate outcome [222]. In the last years, neurofilaments (neurofilament light chain, NFL, and phosphorylated neurofilament heavy chain, pNFH) were under the spotlight for their performance as diagnostic and prognostic biomarkers in the ALS/FTD spectrum (Table 3). Being a structural component of axons, an increase in neurofilament levels in CSF and blood likely reflects ongoing axonal injury [223]. NFLs in CSF and plasma are higher in ALS/FTD spectrum compared to controls [224]. Furthermore, in both ALS and FTD, NFL level rises months before disease onset, and tends to reach a plateau over time [225,226,227]. Recently, NFLs were used in clinical trials as markers of response to treatment. For example, in the phase three clinical trial of the antisense oligonucleotide tofersen for SOD1-ALS (see below), a reduction in plasma NFL levels preceded a significant clinical efficacy at 12-month extension, although no improvement in the primary outcome at 6 months was observed [228]. This result supports the idea that the use of biomarkers as surrogate outcomes might help in detecting biological effects. Moreover, since NFLs rise months before symptom onset, they may be useful in shortening the diagnostic delay in ALS and facilitating an early inclusion in clinical trials [226,229]. Another common neurodegenerative marker, albeit with less evidence, is microtubule-associated protein 2 (MAP2) that has recently been shown to be elevated in ALS CSF [230]. In FTD, recent reports showed that glial fibrillary acidic protein (GFAP), a marker of astrogliosis, was increased only in specific subgroups of FTD patients, discriminating FTD associated with tau (FTD-tau) to FTD-TDP [231] and in symptomatic GRN mutation carriers [232].

Since dysregulation of neuroinflammatory mechanisms is crucial in ALS/FTD pathophysiology [14,233], different immune factors, cells, and pathways are under evaluation as potential biomarkers. For example, many fluid biomarkers of glial activation have been measured, such as soluble triggering receptor expressed on myeloid cells 2 (TREM2) and macrophage-derived chitinases, including chitotriosidase (CHIT1), and YKL-40 (otherwise known as chitinase-3-like protein 1 or CHI3L1). Notably, their role in FTD is still unclear, but in some studies, the chitinase proteins are increased in the CSF of ALS/FTD patients [234,235,236]. Moreover, cytokines and chemokines produced by glial cells have been measured in FTD cohorts, with contrasting results. For example, MCP-1, a proinflammatory cytokine, increased in the CSF of FTD, but RANTES, another proinflammatory cytokine, was reduced in the same cohort [237]. In addition, in ALS, the results are often conflicting, probably related to a rapid change in the inflammatory balance in the different disease phases [14]. Another interesting recent study reported systemic differences in subgroups of ALS patients, with systemic elevation of senescent and late-memory T and B lymphocytes in ALS fast progressors and bulbar patients [201]. In addition, several recent studies have tried to define a protein-profiling characterization aiming to identify disease-specific protein alteration or specific pathology-based mechanisms [238]. For example, with regard to synaptic and neurotransmitter function, none of the synaptic proteins were altered in FTD patients compared to controls [239]. However, a recent study evaluated neuronal pentraxin as a synaptic-dysfunction marker and reported a decrement in the neuronal pentraxin-2 (NPTX2) in *GRN* and *C9ORF72* mutation carriers but not in *MAPT* mutation carriers [234] In conclusion, it is clear from these studies that multiple biomarker panels will presumably be necessary to explore the specific pathogenesis of ALS/FTD and to provide a personalized approach to outcome measures in trials. 

The growing role of Stathmin-2 (STMN-2) in ALS should be described, resulting in one of the missing pieces to the puzzle of TDP-43 proteinopathy. In fact, two independent groups [240,241] recently found that STMN-2 is one of the most abundant transcripts in induced pluripotent stem cell (iPSC)-derived motor neurons, and its expression is regulated by TDP-43. Lower levels of STMN-2 were reported in ALS patients, except for patients mutated for SOD1 [242]. Currently, STMN-2 is still not used as a disease marker in ALS clinical practice, but further investigations are ongoing in this direction. Similar results were obtained for FTD, where truncated STMN2 seems to be a marker for TDP-43 dysfunction in FTD (but not for other pathological phenotypes, such as FTLD-tau) [243]. 

Finally, TDP-43 as a biomarker deserves a dedicated comment. In fact, since TDP-43 pathology accounts for the majority of ALS and for around 45% of FTD, recently a real-time quaking-induced conversion reaction (RT-QuIC) was proposed, disclosing good diagnostic performance to detect TDP-43 aggregates in CSF [244,245]. A recent meta-analysis [246] evaluated the usefulness of CSF TDP-43 as a biomarker of ALS by analyzing 7 studies and including roughly 250 ALS patients. The study reported that CSF TDP-43 was significantly increased in ALS patients compared with controls (Cohen effect: 0.66). However, currently, the included studies were highly heterogenous for sample size, the analytical assays, and patients’ inclusion and exclusion criteria [246]. Therefore, a biomarker directly reflecting TDP-43 dysfunction might be useful to prove target engagement, as well as to facilitate early recruitment in clinical trials, since this event occurs before clinical onset [247], but more studies are required in this regard. 

**Table 3 biomedicines-11-01599-t003:** The most significant biomarkers under study for ALS and FTD. The table shows the main biomarkers that can be used in the ALS/FTD spectrum. Although none are yet used in clinical practice, of these, only neurofilaments are currently used as biomarkers in major clinical trials. Abbreviations: CSF, cerebrospinal fluids, GRN, progranulin.

Biomarker	Patients	Notes	References
**Neurofilaments**	ALS/FTD	Correlation with axonal injury, pathophysiology, and disease progression rate (their levels correlate with shorter survival and more aggressive disease phenotypes); possible diagnostic and treatment markers (i.e., outcome in VALOR trial for SOD1 patients)	[222,223,224,225,226,227,228,229]
**MAP2**	ALS	Increased CSF levels; possible motor neuron degeneration and disease-characterization marker	[230]
**GFAP**	FTD	Raised concentrations in GRN-related FTD; identification of different subgroups of FTD patients; astrogliosis marker; potential marker of proximity to onset	[231,232]
**TREM2, CHIT1, YKL-40**	ALS/FTD	Increase in FTD forms associated with ALS; possible neurodegeneration and neuroinflammation markers in FTD	[234,235,236]
**NPTX2**	GRN and C9orf72 mutation carriers	Reduced levels in patients	[234]
**STMN-2**	ALS/FTD	Lower levels reported in post-mortem brain and spinal cord tissues of familial and sporadic ALS patients; possible diagnostic marker (not yet used as a marker in clinical trials)	[240,241,242,243]
**TDP-43**	ALS/FTD	increased CSF levels; possible target engagement marker (not yet used as a marker in clinical trials)	[246,247]

### 5.2. Experimental Therapies and Clinical Trials

The wide heterogeneity in ALS/FTD clinical and pathogenic mechanisms has led to testing several therapeutic approaches, including those that act on oxidative stress, excitotoxicity, nucleocytoplasmic transport, and neuroinflammation [71]. However, recent evidence has underlined that a single therapeutic approach can hardly be effective in ALS patients [248]. In this context, for ALS, riluzole remains the only approved disease-modifying drug in most European countries. This drug has antiglutamatergic effects and prolongs mean patient survival by 6–19 months [249,250]. After years of failed clinical trials, in 2017, a phase III randomized, double-blind study of intravenous edaravone 60 mg/day showed, in selected ALS patients, a slower reduction in ALSFRS-R after six months of treatment [251]. However, this study has been criticized for its small size, duration, possible adverse events, and lack of data on survival [252]. For these reasons, edaravone has currently been approved for ALS treatment only in the United States, Canada, Japan, South Korea, and Switzerland, but not in the European Union. More recently, the CENTAUR trial [253] demonstrated the efficacy of the sodium phenylbutyrate-taurursodiol (PB-TURSO) in slowing down ALS progression. Sodium phenylbutyrate is a histone deacetylase inhibitor that is involved in targeting signals in mitochondria and endoplasmic reticulum, while taurursodiol (also known as ursodoxicoltaurine) appears to decrease apoptosis. The PB-TURSO, a combination of both agents, reduces neuronal cell death. This agent combination was conditionally approved for use in Canada in June 2022 and approved by the United States Food and Drug Administration (FDA) for all patients with ALS in September 2022. Another promising ongoing clinical trial is for masitinib, a tyrosine kinase inhibitor targeting macrophages, mast cells, and microglia cells, which has immunomodulatory properties. In a double-blind study randomly assigning roughly 400 ALS patients, masitinib showed a 27% in slowing down functional decline compared to placebo [254], and a confirmatory phase III study is ongoing (NCT03127267). A clinical trial with reldesemtiv was recently suspended due to ineffectiveness. Reldesemtiv is a second generation fast skeletal muscle troponin activator with limited penetration of the blood–brain barrier to minimize off-target effects. In a phase II, double-blind, randomized, dose-ranging trial in a large cohort of ALS patients, it did not reach statistical significance [255]. However, a post hoc analysis pooling all active reldesemtiv-treated patients showed trends favoring reldesemtiv as providing some benefit in functional and respiratory functions [256]. For this reason, a phase III trial (COURAGE-ALS, NCT04944784) was started. However, in April 2023, Cytokinetics decided to discontinue the trial evaluating reldesemtiv due to futility. 

Regarding emerging experimental therapies targeted at specific genetic mutations, antisense oligonucleotide (ASO) therapy has been used for genetic forms of ALS; in a phase III trial, a total of 72 participants predicted to have faster progression received tofersen or placebo. Tofersen reduced concentrations of SOD1 in CSF and of neurofilament light chains in plasma than placebo, even without strong clinical benefit in a short follow-up [228]. Based on these biological effects, the tofersen eligibility has been expanded for an early access program to all people with SOD1-ALS, in countries where such programs are permitted by local regulations and access may be secured (https://www.biogen.com/science-and-innovation/access-programs.html, accessed on 22 April 2023). Furthermore, currently, a phase three ATLAS study is ongoing to evaluate whether tofersen can delay clinical onset when initiated in presymptomatic individuals with a *SOD1* genetic mutation and biomarker evidence of disease activity (NCT04856982). In contrast to promising results in *SOD1* mutation carriers, ASO therapy in *C9ORF72* ALS (BIIB078) did not meet any secondary efficacy endpoints or demonstrate clinical benefit (NCT04288856), so the open label extension trial was stopped. For FUS-ALS patients, an ongoing clinical trial aims to investigate the clinical efficacy of ION363 on clinical function and survival (NCT04768972). 

Conversely, in regard to treatment with immunomodulatory drugs, a recent concluded trial has tested the effects of low-dose interleukin-2 in ALS patients, identifying a dose-dependent increase in Treg markers at the end of the treatment period, with concomitant alteration and inhibition of inflammatory pathways [257]. Similar interesting results were also obtained by the Appel group, which demonstrated how Treg/IL-2 treatments were safe and well tolerated and, in a subgroup of patients, also able to slow down disease progression [188]. 

The stem cell treatment for ALS has always been a “hot topic”, albeit with contrasting results over the years. Recently, we discussed the results of a large clinical trial using bone marrow-derived mesenchymal stem cells (MSCs) in ALS. In detail, in a phase II study, these cells were induced into MSCs secreting neurotrophic factors (MSC-NTF) cells to secrete high levels of multiple NTF. Aside from its safety endpoint, the rate of disease progression (ALS functional rating scale-revised slope) had improved at early time points only in fast progressing patients [258]. Although this result may be encouraging, there are still many doubts related to this therapy, such as the cell type, the route and number of administrations, and the appropriate disease stage. A recent post hoc analysis on the long-term effects of MSC transplantation in the CNS of ALS patients described a significantly longer survival in ALS transplanted patients compared to what is expected through the use of the ENCALS model [259]. From a methodological point of view, the result is of great scientific interest, emphasizing a possible long-term effect rather than the immediate post-transplant effect and the possibility of reinvesting in new cell clinical trials. 

As discussed for biomarkers, various strategies for developing treatments targeting TDP-43 are underway. Some researchers are exploring compounds that can prevent the formation of TDP-43 inclusions and subsequent motor neuron death [260]. Others are investigating ways to promote the clearance of abnormal TDP-43 aggregates from affected neurons. This may involve using compounds that stimulate the autophagy [261], the ubiquitin proteasome system [262], and/or the endosomal-lysosomal pathway [140]. Other research has been conducted to stabilize TDP-43 structure and thus, prevent its aggregation. One example is molecular chaperones, which can stabilize the structure of TDP-43 and prevent its misfolding [263]. Gene therapy approaches are being explored to restore normal TDP-43 expression or reduce the protein level of TDP-43 protein [264]. Some recent studies have also focused on immunotherapy, using antibodies or other immune-based approaches to target and clear abnormal TDP-43 aggregates [265]. 

For FTD, currently, there are no effective disease-modifying treatments. Therefore, pharmacologic and nonpharmacologic interventions are aimed at ameliorating mainly the behavioral symptoms of FTD. Nowadays, most therapeutic trials target autosomal dominant forms of FTD, including *C9ORF72* repeat expansions, *GRN* mutations, or *MAPT* mutations. Worthy of mention is the ongoing trial INFRONT-3, a phase three double-blind, placebo-controlled study evaluating the AL001 efficacy and safety in participants at risk for or with FTD due to heterozygous mutations in the progranulin gene (NCT04374136). Few trials are targeting sporadic forms of FTD with tau pathology, acting on different mechanisms, including the enhancement of tau clearance, suppression of toxic tau molecules or its production, alteration of mRNA splicing, and augmentation of tau post-translational modifications [266]. Tau can be targeted with different strategies, such as anti-tau antibodies, tau vaccine, or aggregation inhibition. The studies are still very preliminary and without solid results [267]. Regarding stem cells, although already preliminary tested in AD (with positive results, including improvement in cognitive function and hippocampal tropism) [268], MSCs are not yet tested in FTD. This may depend on multiple factors, including this disorder’s recent proper clinical classification, rarity, and phenotypic variability. 

## 6. Conclusions

With current neuron-centered therapies, only modest progress has been made in ALS and no progress in FTD. In addition, if we consider that the overwhelming majority of cases are sporadic, specific gene-targeting therapies (if successful) will be limited to a minority of cases. It is thus important to target the converging pathways. Proteinopathy and neuroinflammation are common early denominators of a broad array of neurodegenerative diseases, regardless of their etiology, which is exceptionally complex and results from still insufficiently understood interactions between inherited and environmental factors. The breadth of distinct immunopathologies and lack of confirmed (immune)biomarkers is still precluding a straightforward classification of immune defects. For obvious reasons, this also hampers the design of targeted therapies. Nonetheless, the information provided in this work supports the view that the immune component could be crucial in both sporadic and familial disease cases. Interestingly, recent data, such as GWAS enrichment in FTD and immune-linked mutations in ALS/FTD, support the view that dysregulation of the immune system could act not just as a disease modifier, but also as a disease trigger. Importantly, although in ALS the major clinical symptoms arise from the neurodegeneration of motor neurons, and the role of CNS-resident microglia in the ALS pathogenesis has been well established, recent evidence also clearly shows that immune alterations are not limited to CNS. Finally, with so many shared genetic and environmental factors across a wide spectrum of neurodegenerative diseases, it is puzzling why some individuals may develop either ALS, AD, or PD. This all leads us to conclude that genetic, epidemiological, neurobiology, and toxicology studies aimed at identifying risk factors, mechanisms, and treatments may initially target one disease, but may also collectively advance the understanding of other neurodegenerative diseases. Nevertheless, although common targets are of particular interest, detailed patient profiling is essential since therapeutic approaches will likely have to be carefully customized to underlying genetics, neuropathology, immunopathology, and the disease stage. 

## Figures and Tables

**Figure 1 biomedicines-11-01599-f001:**
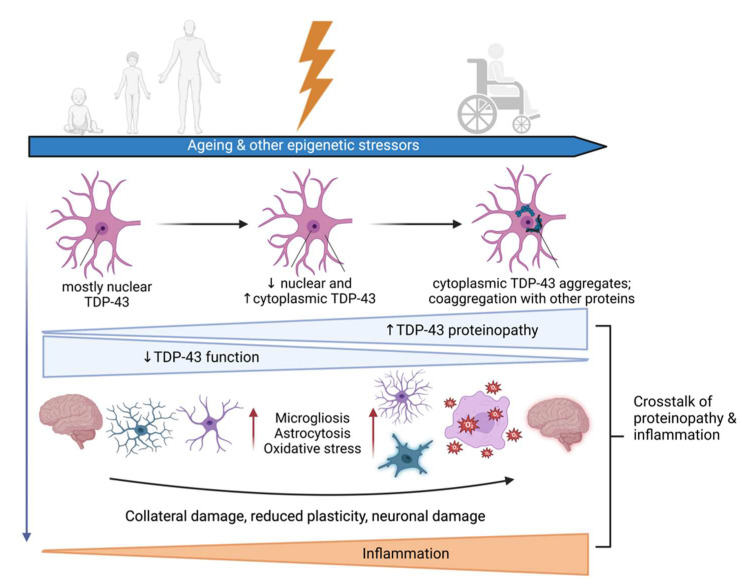
Progressive increase in TAR DNA-binding protein 43 (TDP-43) proteinopathy in amyotrophic lateral sclerosis (ALS) and frontotemporal degeneration (FTD) results in pathology in neurons and glia. Both loss- and gain-of-function TDP-43-mediated events contribute to the pathology. During this process, TDP-43 co-aggregates with other cellular proteins and TDP-43 pathology crosstalks to inflammation. These events are highlighted, and further discussed in the text. Created by BioRender.com.

**Figure 2 biomedicines-11-01599-f002:**
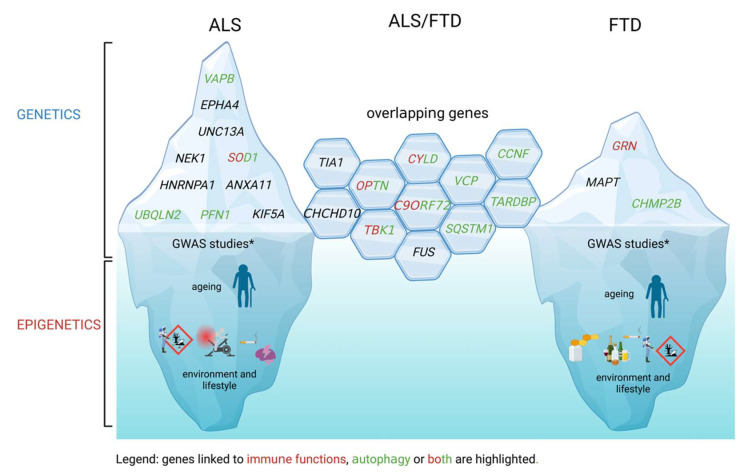
Amyotrophic lateral sclerosis (ALS) and frontotemporal degeneration (FTD): direct and indirect links between genes and immunity. ALS and FTD, similar to other adult-onset neurodegenerative diseases, have both genetic (shown as the tip of the iceberg) and epigenetic (shown as the bottom of the iceberg) backgrounds. Strong mendelian genes are depicted and divided into those linked only to ALS, only to FTD or both. The genes directly implicated in immune functions (immune signaling, reactive oxygen species (ROS) production) are highlighted in red, whereas those linked to autophagy are highlighted in green; those implicated in both autophagy and immune signaling are red/green. Although the epigenetics factors such as ageing and diverse environmental risk factors (smoking, exposure to environmental toxins, and alcohol) play an important role in ALS and FTD development, their exact impact on disease progression is still unknown. * Multiple genome-wide association studies (GWAS) studies have also shown linkages whose direct effects need to be clarified. Created by BioRender.com.

**Figure 3 biomedicines-11-01599-f003:**
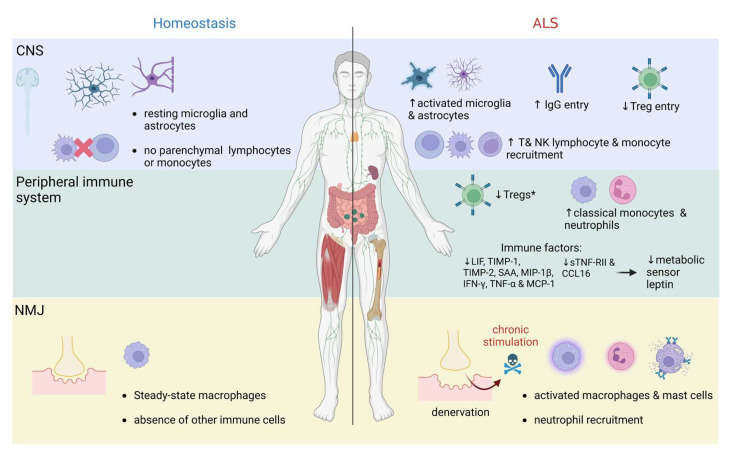
ALS is a systemic disorder. Immune system dysregulation in amyotrophic lateral sclerosis (ALS) is present at multiple levels: (A) central nervous system (CNS), seen as activated microglia and astrocytes; increased IgG, T cell memory (Tm) and natural killer (NK) cell entry, and decreased Treg entry; remark: the extent of monocyte recruitment is still controversial, (B) peripheral immune system is represented here in the blood (various blood markers have been highlighted including an increase in classical monocytes, neutrophils, Th1, and Th17 CD4+ T cells; decrease in Treg, especially in fast progressors; however, no major increase in proinflammatory cytokines was pinpointed, whereas some studies found decreased inflammatory and metabolic factors), and (C) neuromuscular junction (NMJ), where an increase in activated macrophages and mast cells, and neutrophil infiltration are observed. * Findings correlated to fast disease progression. Created by BioRender.com.

**Figure 4 biomedicines-11-01599-f004:**
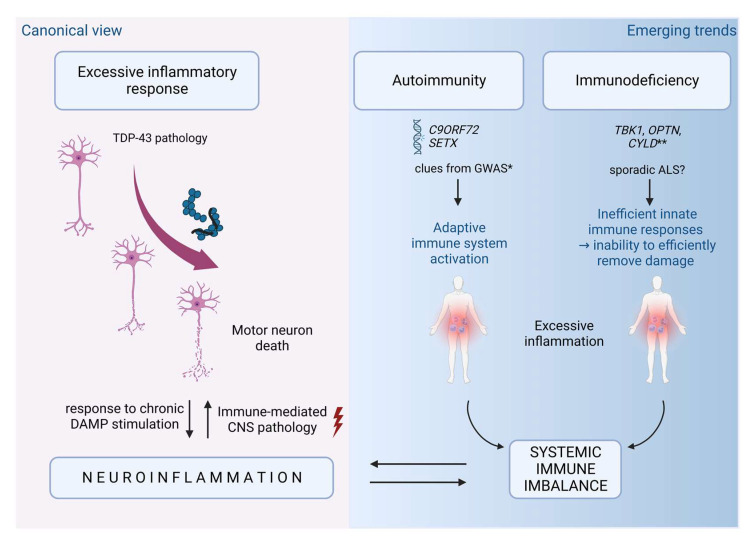
Causes of immune imbalance in amyotrophic lateral sclerosis (ALS) and frontotemporal degeneration (FTD). The excessive inflammatory response is a direct consequence of the protracted inability to efficiently clean up protein aggregates or neuronal debris, representing the outcome that is commonly linked to ALS and FTD. The resulting inflammatory response can, in turn, become the major cause of neuronal death, and further increase TAR DNA-binding protein (TDP-43) proteinopathy, resulting in a vicious cycle. Two other emerging scenarios are immunodeficiency and autoimmunity. They are linked to specific genetic mutations exemplified by the role of *C9ORF72* and senataxin (*SETX*) in autoimmunity, and *TBK1*, *OPTN*, and *CYLD* in immunodeficiency. Autoimmunity and immunodeficiency could also be triggered by ageing and other environmental risk factors in sporadic ALS patients. Sporadic ALS patients show a complex immune profile that can include a decrease in several inflammatory markers. Excessive inflammation linked to autoimmunity and immunodeficiency could potentially be a primary disease trigger, thus representing an early step in disease pathogenesis. * Clues from genome-wide association studies (GWAS) studies link autoimmune diseases to FTD but not to ALS; ** Given that *TBK1* and *OPTN* are multifunctional proteins, their mutations could affect non-immune functions as well. Created by BioRender.com.

**Table 1 biomedicines-11-01599-t001:** ALS/FTD: genes and mechanisms. Mendelian genes associated with amyotrophic lateral sclerosis (ALS), frontotemporal degeneration (FTD) and ALS/FTD are shown. Cellular processes affected by gene dysfunction that are potentially relevant for disease pathogenesis are listed for each gene.

Disease	Genes	Affected Processes	References
**ALS**	*VAPB*	autophagy, RNA binding, protein homeostasis, mitochondrial functions, vesicle trafficking	[65,66,67]
	*EPHA4*	motor neuron survival	[68]
	*UNC13A*	neurotransmission	[69]
	*NEK1*	RNA metabolism, DNA repair, axonal polarity, neuronal morphology	[70]
	*SOD1*	autophagy, mitophagy, RNA metabolism, protein homeostasis, mitochondrial and immune functions	[71,72,73]
	*HNRNPA1*	protein folding, stress granule dynamics	[74]
	*ANXA11*	calcium homeostasis, stress granule dynamics, axon morphology	[71,75]
	*PFN1*	autophagy, RNA metabolism, stress granule dynamics	[76,77]
	*KIF5A*	trafficking and neuronal homeostasis	[78]
	*UBQLN2*	autophagy, RNA metabolism	[50,71,79]
**ALS/FTD**	*VCP*	autophagy, mitochondrial function	[80,81]
	*OPTN*	autophagy, mitophagy, vesicular trafficking, immune signaling	[37]
	*CHCHD10*	mitochondrial function	[82]
	*C9ORF72*	autophagy, RNA metabolism, protein homeostasis, nucleocytoplasmic transport	[73,83,84,85]
	*TBK1*	autophagy, mitophagy, protein homeostasis, mitochondrial function	[86,87]
	*TARDBP*	autophagy, nucleocytoplasmic transport, RNA metabolism, axonal transport	[88,89,90]
	*CYLD*	autophagy, immune signaling	[10,91]
	*FUS*	nucleocytoplasmic transport, DNA damage repair, RNA metabolism	[71]
	*SQSTM1*	autophagy	[92]
	*CCNF*	autophagy, axon morphology	[71,93]
	*TIA1*	stress granule dynamics	[94]
**FTD**	*GRN*	immune signaling, lysosomal functions	[95]
	*MAPT*	vesicular trafficking, lysosomal functions	[96]
	*CHMP2B*	autophagy	[97]

**Table 2 biomedicines-11-01599-t002:** Most significant proteins present in the aggregates in ALS/FTD spectrum. Abbreviations: TDP-43, TAR DNA-binding protein 43; AD, Alzheimer’s disease; CBD, corticobasal degeneration; MSA, multiple system atrophy; FTD, frontotemporal degeneration; mRNP, messenger ribonucleoprotein; ALS, amyotrophic lateral sclerosis; PD, Parkinson’s disease; PSF: protein-associated splicing factor; VCP: valosin-containing protein.

Co-Aggregating Proteins and Peptides	Notes	References
**Amyloidogenic proteins**	Using an anti-oligomer antibody, potential hybrid oligomers composed of amyloid-β, prion protein, α-synuclein, and TDP-43 phosphorylated at serine 409/410 were detected in AD brains. Colocalization of α-synuclein, tau, and TDP-43 has also been occasionally reported in patients suffering from CBD and MSA.	[115,116,117]
**ATXN2**	ATXN2 and TDP-43 colocalize in cytoplasmic inclusions in FTD. ATXN2 and TDP-43 associated in a complex that is dependent on RNA and can act as a powerful disease modifier. In a mouse TDP-43 model, the decrease in ataxin-2 markedly increased survival and improved motor function.	[118,119,120]
**C9orf72 DPRs**	TDP-43 has been shown to colocalize with poly-GR and poly-PA inclusions. No colocalization was observed for poly-GP, poly-GA, or poly-PR immunoreactive inclusions. In the motor regions of C9 ALS cases, only poly-GR dendritic aggregations had significant colocalization with phosphorylated TDP-43.	[121,122]
**CDK5**	CDK5-positive granules have been shown to overlap with pSmad2/3, ubiquitin, and phospho-TDP-43 in several AD patients.	[123]
**DISC1**	Cytosolic TDP-43 and DISC1 co-aggregate in brains of both FTD mouse models and FTD patients and disrupt the activity-dependent local translation in dendrites.	[124]
**HuD/ELAVL4**	ELAVL4 has been found as a neural-specific component of FUS-positive cytoplasmic aggregates, whereas in sporadic ALS patients, it colocalized with positive inclusions of phosphorylated TDP-43.	[125]
**ERp57**	ERp57 colocalizes with phospho-TDP-43-positive inclusions present in sporadic ALS patients.	[126]
**GPNMB**	GPNMB aggregates colocalize with TDP-43 in the spinal cord of ALS patients. In NSC-34 cells, the expression level of GPNMB increased by overexpression of mutant M337V and A315T TDP-43.	[127]
**hnRNP E2**	hnRNP E2 immunostaining colocalizes with TDP-43 pathological changes, but only in patients with semantic dementia and FTD type C TDP-43 histology.	[128]
**HERV-K RT**	The reverse transcriptase protein of this endogenous retrovirus was observed to localize to cortical neurons of ALS patients and strongly correlated with TDP-43 expression.	[129]
**IL-10**	IL-10 colocalizes with TDP-43-positive cytoplasmic inclusions in anterior horn motor neurons in ALS patients.	[130]
**Nup62**	Cytoplasmic NUP62-TDP-43 inclusions are frequently found in C9orf72 ALS/FTD as well as in sporadic ALS/FTD post-mortem CNS tissue.	[131]
**OPTN**	Mutations in the *OPTN* gene have been reported to be causative for familial ALS and FTD, but mutated optineurin has not been found in aggregates. In contrast, in sALS cases optineurin has been observed in cytoplasmic skein-like inclusions colocalizing with ubiquitin, TDP-43, and possibly FUS; similar optineurin positive inclusions have been reported in AD, PD, Creutzfeldt-Jakob and Pick disease.	[132,133]
**p62/SQSTM1**	p62 physiologically binds to TDP-43 and is involved in degradation of the TDP-43 35-kDa fragment. It also colocalizes with TDP-43-positive cytoplasmic inclusions, ubiquitin, and UBQLN2 in patients with FTD/ALS.	[134,135]
**PABP-1**	This protein colocalizes to mature TDP-43 inclusions (but not to pre-inclusions) in ALS motor neurons and is more prevalent in patients bearing *C9ORF72* expansions.	[136]
**PFN1**	Profilin mutations can induce aggregation of TDP-43 and PFN1 in inclusions positive for phosphorylated TDP-43 in ALS patients.	[137,138]
**RBM14, PSF, NONO**	These paraspeckle markers are found in insoluble TDP-43 artificial aggregates together with stress granule markers. In a proteomic study, PSF was found to be enriched in the TDP-43-positive detergent-insoluble proteome of four post-mortem FTD patients.	[139]
**Rab5**	In yeast, TDP-43 foci colocalized frequently with endogenous Rab5 foci, suggesting a greater association of TDP-43 with endosomal-like compartments over autophagic compartments.	[140]
**RBM45**	RBM45 colocalizes with TDP-43 in inclusion bodies and is especially present in ALS/FTD patients with *C9ORF72* expansions. It was later shown that RBM45 forms homo-oligomers and physically associates with TDP-43 and FUS in the nucleus.	[141,142,143]
**RGNEF**	Mutations in the murine homologue of this protein cause altered NFL mRNA stability and lead to NF aggregate formation and motor neuronopathy. This protein can also interact by immunoprecipitation with FUS and p62. Mutations in *RGNEF* have been described in an ALS family.	[144,145,146]
**RANGAP1** **/NUP205**	Aberrant colocalizations of TDP-43 with the nuclear pore complex proteins RanGAP1 and NUP205 have been observed in motor neurons of ALS patients.	[147]
**Tau**	Occasional colocalization of TDP-43 with tau has been described in globular astrocytic inclusions (GAIs) of a Japanese patient affected by ALS/FTD. A recent survey of more than 200 AD patients has observed that almost 30% of TDP-positive cases colocalized with phosphorylated tau (detected using PHF-1 antibody).	[148,149]
**TTBK1/TTBK2**	These kinases have been observed to colocalize with phosphorylated TDP-43 in human post-mortem tissues from both FTD and ALS cases.	[150]
**UBQLN2**	In autopsy material of human spinal cord samples of *UBQLN2* mutation carriers, its skein-like inclusions are positive for UBQLN2, ubiquitin, p62, TDP-43, FUS, and OPTN, but not SOD1.	[151,152]
**VHL/CUL2 E3 complex**	VHL preferentially recognizes misfolded forms of TDP-43 and promotes ubiquitin-mediated proteasomal degradation of fragmented forms of TDP-43. Phosphorylated TDP-43 and VHL are occasionally colocalized in cytoplasmic inclusions in oligodendrocytes in ALS.	[153]
**VCP**	In hippocampal dentate gyrus neurons of *C9ORF72* patients, VCP inclusions have been reported to co-aggregate with phospho-TDP-43.	[63]

## Data Availability

Not applicable.

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
