# Peer review of "Emerging Trends in the Field of Inflammation and Proteinopathy in ALS/FTD Spectrum Disorder"

_biomedicines, 2023, doi:10.3390/biomedicines11061599_

Round 1

Reviewer 1 Report

The review by Marchi et al., "Emerging trends in the field of inflammation and proteinopathy in ALS/FTD spectrum and related disorders" highlight the emerging trend in amyotrophic lateral sclerosis. The review is written well and looks nice.

Author Response

We thank the reviewer for kind comments. 

Reviewer 2 Report

Overall, the authors recount the emerging trends in ALS and FTD and predominantly focus on neuroinflammation and systemic immune imbalance in ALS and FTD. I have some questions.

1.     A table about 2.1. Mendelian genetic elements, with necessary information about gene or relationship with ALS/FTP, is recommended. And the current table 1 should be further summarized.

2.     Also about 2.1. Mendelian genetic elements, is there any connection in these genes?  For instance, Optineurin is also regulated by TBK1 during autophagy.

3.     A more detailed mechanism of inflammation related molecules in 4. Immunity in ALS and FTD should be added.

4.     Also a table about 5.1. Biomarkers should be added to introduce the progress about biomarkers. Besides, the targets and progress mentioned in 5.2 Experimental therapies and clinical trials also should in included in this table, and connected with biomarkers.

5.     I think the part of 6. Related brain disorders should be deleted.

Author Response

We thank the reviewer for helpful suggestions. Our point-by-point reply is below:

1. "A table about “2.1. Mendelian genetic elements”, with necessary information about gene or relationship with ALS/FTD, is recommended. And the current table 1 should be further summarized."

As recommended by the reviewer, we have added a table for chapter 2.1. in Mendelian genetic elements chapter, and have summarised the table 1 in Proteinopathy chapter (now table 2). We have also updated the figure 2 to provide better links to the chapter and the new table. 

2. "Also about “2.1. Mendelian genetic elements”, is there any connection in these genes? For instance, Optineurin is also regulated by TBK1 during autophagy."

We thank the reviewer for pointing this out. To address this, we have now added the following sentence (with the relevant references): The crosstalk between these proteins has also been reported: for example, TBK1, p62 and optineurin partner in the same molecular pathways: TBK1 is important for phosphorylation of optineurin and p62 during autophagy, and optineurin regulates TBK1 cell localization and activation during immune signaling [54–57]. 

3. "A more detailed mechanism of inflammation related molecules in “4. Immunity in ALS and FTD” should be added."

As recommended by the reviewer we have updated the text and figure 3. The changes include:

Depending on the disease stage, activated glia can exert various antiinflammatory (secrete cytokines IL-10 and Il-4, phagocytose debris, increase damage sensor expression) and proinflammatory functions (secrete cytokines, such as tumor necrosis factor (TNF), IL-1β IL-6, chemokines CCL2 and IL-8, increase damage sensor expressions, generate  ROS and nitric oxide) (reviewed in: Beers and Appel, 2019; Beland et al., 2020).

Figure 3. ALS is a systemic disorder. Immune system dysregulation in amyotrophic lateral sclerosis (ALS) is present at multiple levels: A) central nervous system (CNS), seen as ac- tivated microglia and astrocytes; increased IgG, T and natural killer (NK) cells entry, and decreased Treg entry; remark: the extent of monocyte recruitment is still controversial, B) peripheral immune system, represented here in blood (various blood markers have been highlighted including increase in classical monocytes, neutrophils, Th1 and Th17 CD4+ T cells; decrease in Treg, especially in fast progressors; however, no major increase in pro- inflammatory cytokines was pinpointed, whereas some studies found decreased inflam- matory and metabolic factors), and C) neuromuscular junction (NMJ), where an increase in activated macrophages and mast cells, and neutrophil infiltration are observed. *Findings correlated to fast disease progression

The gene profiling studies performed on the blood monocytes from
ALS patients revealed a molecular signature of chronic activation through lipopolysaccharide (LPS)-Toll-like receptor (TLR)2/4 signaling pathway suggesting a systemic deregulation of innate immune response [160], although without an overt increase in blood proinflammatory markers [166,167], although more studies, especially longitudinal ones are
of crucial importance. 

4. "Also a table about “5.1. Biomarkers” should be added to introduce the progress about biomarkers. Besides, the targets and progress mentioned in “5.2 Experimental therapies and clinical trials” also should in included in this table, and connected with biomarkers."

As recommended by the reviewer, we have added a table in the Clinical studies, biomarkers and emerging therapies in ALS/FTD chapter.  

5. "I think the part of “6. Related brain disorders” should be deleted." 

This was a tough call for us because we wanted to spread the message on the interrelatedness of neurodegenerative diseases (and have gathered to this end experts on AD, PD and NPC,  but since the same questions has been raised by reviewer #3, we have removed this chapter. 

Reviewer 3 Report

 In this review, the authors provide a comprehensive summary of the emerging trends in amyotrophic lateral sclerosis (ALS) and frontotemporal disease (FTD), focusing on neuroinflammation, immune system and proteinopathy (mainly TDP-43). Specially, the authors present in a table and discuss the most significant proteins present in aggregates in these both diseases. The review also highlights recent discoveries in experimental models and clinical trials. This review is of exceptional quality and provides valuable insights into the topic of ALS and FTD, however some minor points should be address:

1.     Figure 3 should be explained in detail in the legend.

2.     For all figures, the legend should be after the imagine.

3.     Authors should considered add a table in which resume the biomarkers and their importance in ALS and FTD.

4.     It is not clear section 6, the authors should add a paragraph linking this section to the rest of the text, because I cannot understand it relevance in the review.

5.     Authors add to many abbreviations without add their meaning, please add their meaning.

Author Response

We thank the reviewer for kind comments and helpful suggestions. Our point-by-point reply is below:

1. "Figure 3 should be explained in detail in the legend."

As recommended by the reviewer, we have further elaborated the legend and it now reads: ALS is a systemic disorder. Immune system dysregulation in amyotrophic lateral sclerosis (ALS) is present at multiple levels: A) central nervous system (CNS), seen as ac- tivated microglia and astrocytes; increased IgG, T and natural killer (NK) cells entry, and decreased Treg entry; remark: the extent of monocyte recruitment is still controversial, B) peripheral immune system, represented here in blood (various blood markers have been highlighted including increase in classical monocytes, neutrophils, Th1 and Th17 CD4+ T cells; decrease in Treg, especially in fast progressors; however, no major increase in pro- inflammatory cytokines was pinpointed, whereas some studies found decreased inflam- matory and metabolic factors), and C) neuromuscular junction (NMJ), where an increase in activated macrophages and mast cells, and neutrophil infiltration are observed. *Findings correlated to fast disease progression

2. "For all figures, the legend should be after the image."

We have corrected the order. 

3.  "Authors should considered add a table in which resume the biomarkers and their importance in ALS and FTD."

As recommended by both reviewers, we have added a table in the Clinical studies, biomarkers and emerging therapies in ALS/FTD chapter. 

4. "It is not clear section 6, the authors should add a paragraph linking this section to the rest of the text, because I cannot understand it relevance in the review."

Given the similar comments by reviewers #2 and #3, we have removed this chapter. Hopefully, we will manage to make our message on overlapping points between neurodegenerative diseases more clear in another publication :)

5. "Authors add to many abbreviations without add their meaning, please add their meaning."

We thank the reviewer for noticing this. We have added the missing explanations. Finally, we also made the following abbreviation paragraph:

Abbreviations: amyotrophic lateral sclerosis (ALS); frontotemporal disease (FTD); TAR DNA-bind- ing protein 43 (TDP-43); frontotemporal degeneration (FTD); central nervous system (CNS); genome wide association studies (GWAS); reactive oxygen species (ROS); repeat associated non-AUG (RAN); dipeptide repeat proteins (DRP); interleukin (IL); stimulator of interferon genes (STING); superoxide dismutase (SOD1); tumor necrosis factor α (TNF-α); interferon γ (IFN-γ); wild-type (WT); nuclear factor-κB (NF-κB); interferon regulatory factor 3 (IRF-3); Neuro2A (N2A); single-nu- cleotide polymorphisms (SNPs); interleukin 18 receptor accessory protein (IL18RAP); Alzheimer’s disease (AD); corticobasal degeneration (CBD); glial cytoplasmic inclusions (GCis); multiple system atrophy (MSA); messenger ribonucleoprotein (mRNP); PD, Parkinson’s disease (PD); pro-tein-asso- ciated splicing factor (PSF); valosin containing protein (VCP); neuromuscular junction (NMJ); mu- tated human SOD1 (mSOD1); cerebrospinal fluid (CSF); natural killer (NK); AMP-activated protein kinase (AMPK); damage- and pathogen-associated molecular patterns (DAMP/PAMP); amyloid beta (Aβ); human leukocyte antigen (HLA); senataxin (SETX); neurofilament light chain (NfL); phosphorylated neurofilament heavy chain (pNfH); microtubule-associated protein 2 (MAP2); glial fibrillary acidic protein (GFAP); frontotemporal degeneration associated with tau (FTD-tau); soluble triggering receptor expressed on myeloid cells 2 (TREM2); chitotriosidase (CHIT1); neuronal pen- traxin-2 (NPTX2); Stathmin-2 (STMN-2); induced pluripotent stem cell (iPSC); real-time quaking- induced conversion reaction (RT-QuIC); phenylbutyrate-taurursodiol (PB-TURSO); antisense oligo- nucleotide (ASO); mesenchymal stem cells (MSCs); MSCs secreting neurotrophic factors (MSC- NTF); late onset Alzheimer's disease (LOAD); TANK binding kinase 1 (TBK1); lipopolysaccharide (LPS); Toll-like receptors (TLR); leukemia inhibitory factor (LIF); tissue inhibitor of metalloprotein- ase 1 (TIMP-1); tissue inhibitor of metalloproteinase 2 (TIMP-2); serum amyloid A (SAA); macro- phage inflammatory protein-1 beta (MIP-1β); monocyte chemoattractant protein (MCP-1).